# Neural Outlier Rejection for Self-Supervised Keypoint Learning

**Jiexiong Tang**[1,2]   **Hanme Kim**[1]   **Vitor Guizilini**[1]   **Sudeep Pillai**[1]   **Rareş Ambruş**[1]

[1] Toyota Research Institute (TRI)   [2] KTH Royal Institute of Technology

[1]`{first.last}@tri.global`   [2]`jiexiong@kth.se`

## Abstract

Identifying salient points in images is a crucial component for visual odometry, Structure-from-Motion or SLAM algorithms. Recently, several learned keypoint methods have demonstrated compelling performance on challenging benchmarks. However, generating consistent and accurate training data for interest-point detection in natural images still remains challenging, especially for human annotators. We introduce *IO-Net* (i.e. *InlierOutlierNet*), a novel proxy task for the self-supervision of keypoint detection, description and matching. By making the sampling of inlier-outlier sets from point-pair correspondences fully differentiable within the keypoint learning framework, we show that are able to simultaneously self-supervise keypoint description and improve keypoint matching. Second, we introduce *KeyPointNet*, a keypoint-network architecture that is especially amenable to robust keypoint detection and description. We design the network to allow local keypoint aggregation to avoid artifacts due to spatial discretizations commonly used for this task, and we improve fine-grained keypoint descriptor performance by taking advantage of efficient sub-pixel convolutions to upsample the descriptor feature-maps to a higher operating resolution. Through extensive experiments and ablative analysis, we show that the proposed self-supervised keypoint learning method greatly improves the quality of feature matching and homography estimation on challenging benchmarks over the state-of-the-art.[†]

## 1 Introduction

Detecting interest points in RGB images and matching them across views is a fundamental capability of many robotic systems. Tasks such Simultaneous Localization and Mapping (SLAM) (Cadena et al., 2016), Structure-from-Motion (SfM) (Agarwal et al., 2010) and object detection assume that salient keypoints can be detected and re-identified in a wide range of scenarios, which requires invariance properties to lighting effects, viewpoint changes, scale, time of day, etc. However, these tasks still mostly rely on handcrafted image features such as SIFT (Lowe et al., 1999) or ORB (Rublee et al., 2011), which have been shown to be limited in performance when compared to learned alternatives (Balntas et al., 2017).

Deep learning methods have revolutionized many computer vision applications including 2D/3D object detection (Lang et al., 2019; Tian et al., 2019), semantic segmentation (Li et al., 2018; Kirillov et al., 2019), human pose estimation (Sun et al., 2019), etc. However, most learning algorithms need supervision and rely on labels which are often expensive to acquire. Moreover, supervising interest point detection is unnatural, as a human annotator cannot readily identify salient regions in images as well as key signatures or descriptors, which would allow their re-identification. Self-supervised learning methods have gained in popularity recently, being used for tasks such as depth regression (Guizilini et al., 2019), tracking (Vondrick et al., 2018) and representation learning (Wang et al., 2019; Kolesnikov et al., 2019). Following DeTone et al. (2018b) and Christiansen et al. (2019), we propose a self-supervised methodology for jointly training a keypoint detector as well as its associated descriptor.

---

[†]Code: https://github.com/TRI-ML/KP2D

Our main contributions are: (i) We introduce *IO-Net* (i.e. *InlierOutlierNet*), a novel proxy task for the self-supervision of keypoint detection, description and matching. By using a neurally-guided outlier-rejection scheme (Brachmann & Rother, 2019) as an auxiliary task, we show that we are able to simultaneously self-supervise keypoint description and generate optimal inlier sets from possible corresponding point-pairs. While the keypoint network is fully self-supervised, the network is able to effectively learn distinguishable features for two-view matching, via the flow of gradients from consistently matched point-pairs. (ii) We introduce *KeyPointNet*, and propose two modifications to the keypoint-network architecture described in Christiansen et al. (2019). First, we allow the keypoint location head to regress keypoint locations outside their corresponding cells, enabling keypoint matching near and across cell-boundaries. Second, by taking advantage of sub-pixel convolutions to interpolate the descriptor feature-maps to a higher resolution, we show that we are able to improve the fine-grained keypoint descriptor fidelity and performance especially as they retain more fine-grained detail for pixel-level metric learning in the self-supervised regime. Through extensive experiments and ablation studies, we show that the proposed architecture allows us to establish state-of-the-art performance for the task of self-supervised keypoint detection, description and matching.

## 2 RELATED WORK

The recent success of deep learning-based methods in many computer vision applications, especially feature descriptors, has motivated general research in the direction of image feature detection beyond handcrafted methods. Such state-of-the-art learned keypoint detectors and descriptors have recently demonstrated improved performance on challenging benchmarks (DeTone et al., 2018b; Christiansen et al., 2019; Sarlin et al., 2019). In TILDE (Verdie et al., 2015), the authors introduced multiple piece-wise linear regression models to detect features under severe changes in weather and lighting conditions. To train the regressors, they generate *pseudo* ground truth interest points by using a Difference-of-Gaussian (DoG) detector (Lowe, 2004) from an image sequence captured at different times of day and seasons. LIFT (Yi et al., 2016) is able to estimate features which are robust to significant viewpoint and illumination differences using an end-to-end learning pipeline consisting of three modules: interest point detection, orientation estimation and descriptor computation. In LF-Net (Ono et al., 2018), the authors introduced an end-to-end differentiable network which estimates position, scale and orientation of features by jointly optimizing the detector and descriptor in a single module.

Quad-networks (Savinov et al., 2017) introduced an unsupervised learning scheme for training a shallow 2-layer network to predict feature points. SuperPoint (DeTone et al., 2018b) is a self-supervised framework that is trained on whole images and is able to predict both interest points and descriptors. Its architecture shares most of the computation in the detection and description modules, making it fast enough for real-time operation, but it requires multiple stages of training which is not desirable in practice. Most recently, UnsuperPoint (Christiansen et al., 2019) presented a fast deep-learning based keypoint detector and descriptor which requires only one round of training in a self-supervised manner. Inspired by SuperPoint, it also shares most of the computation in the detection and description modules, and uses a siamese network to learn descriptors. They employ simple homography adaptation along with non-spatial image augmentations to create the 2D synthetic views required to train their self-supervised keypoint estimation model, which is advantageous because it trivially solves data association between these views. In their work, Christiansen et al. (2019) predict keypoints that are evenly distributed within the cells and enforce that the predicted keypoint locations do not cross cell boundaries (i.e. each cell predicts a keypoint inside it). We show that this leads to sub-optimal performance especially when stable keypoints appear near cell borders. Instead, our method explicitly handles the detection and association of keypoints across cell-boundaries, thereby improving the overall matching performance. In Self-Improving Visual Odometry (DeTone et al., 2018a), the authors first estimate 2D keypoints and descriptors for each image in a monocular sequence using a convolutional network, and then use a bundle adjustment method to classify the stability of those keypoints based on re-projection error, which serves as supervisory signal to re-train the model. Their method, however, is not fully differentiable, so it cannot be trained in an end-to-end manner. Instead, we incorporate an end-to-end differentiable and neurally-guided outlier-rejection mechanism (IO-Net) that explicitly generates an additional proxy supervisory signal for the matching input keypoint-pairs identified by our KeyPointNet architecture. This allows the keypoint

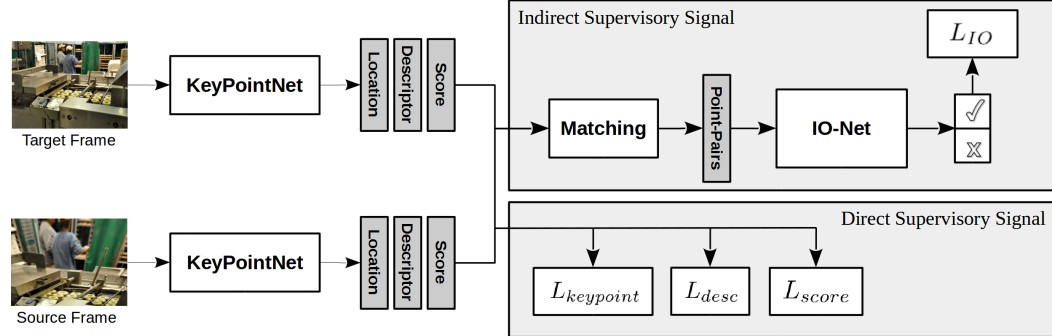

Figure 1: Our proposed framework for self-supervised keypoint detector and descriptor learning using *KeyPointNet* and *IO-Net*. The KeyPointNet is optimized in an end-to-end differentiable manner by imposing an explicit loss on each of the 3 target outputs (score, location and descriptor). Additionally, the IO-Net produces an indirect supervisory signal to KeyPointNet targets by propagating gradients from the classification of matching input point-pairs.

descriptions to be further refined as a result of the outlier-rejection network predictions occurring during the two-view matching stage.

# 3 SELF-SUPERVISED KEYPOINT LEARNING

In this work, we aim to regress a function which takes as input an image and outputs keypoints, descriptors, and scores. Specifically, we define $K : I \rightarrow \{\mathbf{p}, \mathbf{f}, \mathbf{s}\}$, with input image $I \in \mathbb{R}^{3 \times H \times W}$, and output keypoints $\mathbf{p} = \{[u, v]\} \in \mathbb{R}^{2 \times N}$, descriptors $\mathbf{f} \in \mathbb{R}^{256 \times N}$ and keypoint scores $\mathbf{s} \in \mathbb{R}^{N}$; $N$ represents the total number of keypoints extracted and it varies according to an input image resolution, as defined in the following sections. We note that throughout this paper we use $\mathbf{p}$ to refer to the set of keypoints extracted from an image, while $p$ is used to refer to a single keypoint.

Following the work of Christiansen et al. (2019), we train the proposed learning framework in a self-supervised fashion by receiving as input a source image $I_s$ such that $K(I_s) = \{\mathbf{p_s}, \mathbf{f_s}, \mathbf{s_s}\}$ and a target image $I_t$ such that $K(I_t) = \{\mathbf{p_t}, \mathbf{f_t}, \mathbf{s_t}\}$. Images $I_s$ and $I_t$ are related through a known homography transformation $\mathbf{H}$ which warps a pixel from the source image and maps it into the target image. We define $\mathbf{p}_t^* = \{[u_i^*, v_i^*]\} = \mathbf{H}(\mathbf{p}_s)$, with $i \in I$ - e.g. the corresponding locations of source keypoints $\mathbf{p_s}$ after being warped into the target frame.

Inspired by recent advances in Neural Guided Sample Consensus methods (Brachmann & Rother, 2019), we define a second function $C$ which takes as input point-pairs along with associated weights according to a distance metric, and outputs the likelihood that each point-pair belongs to an inlier set of matches. Formally, we define $C : \{\mathbf{p_s}, \mathbf{p_t^*}, d(\mathbf{f_s}, \mathbf{f_t^*})\} \in \mathbb{R}^{5 \times N} \rightarrow \mathbb{R}^{N}$ as a mapping which computes the probability that a point-pair belongs to an inlier set. We note that $C$ is only used at training time to choose an optimal set of consistent inliers from possible corresponding point pairs and to encourage the gradient flow through consistent point-pairs.

An overview of our method is presented in Figure 1. We define the model $K$ parametrized by $\theta_K$ as an encoder-decoder style network. The encoder consists of 4 VGG-style blocks stacked to reduce the resolution of the image $H \times W$ to $Hc \times Wc = H/8 \times W/8$. This allows an efficient prediction for keypoint location and descriptors. In this low resolution embedding space, each pixel corresponds to an $8 \times 8$ cell in the original image. The decoder consists of 3 separate heads for the keypoints, descriptors and scores respectively. Thus for an image of input size $H \times W$, the total number of keypoints regressed is $(H \times W)/64$, each with a corresponding score and descriptor. For every convolutional layer except the final one, batch normalization is applied with leakyReLU activation. A detailed description of our network architecture can be seen in Figure 2. The IO-Net is a 1D CNN parametrized by $\theta_{IO}$, for which we follow closely the structure from Brachmann & Rother (2019) with 4 default setting residual blocks and the original activation function for final layer is removed. A more detailed description of these networks can be found in the Appendix (Tables 6 and 7).

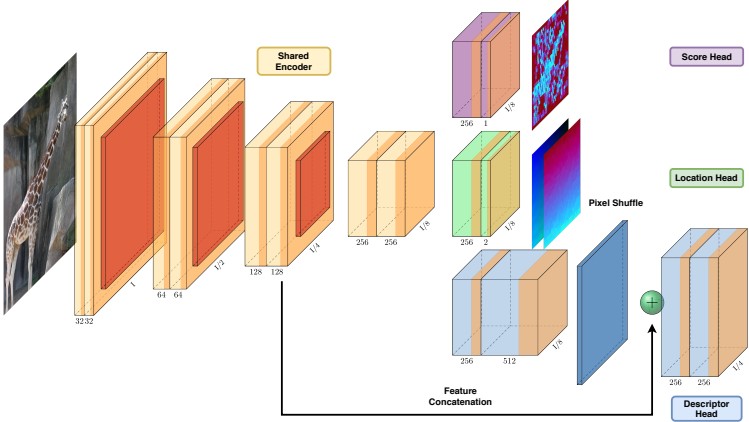

Figure 2: The proposed *KeyPointNet* architecture leverages a shared-encoder backbone with three output heads for the regression of keypoint locations (**p**), scores (**s**) and descriptions (**f**). To further improve keypoint description performance, KeyPointNet produces higher-resolution feature-maps for the keypoint descriptors via efficient sub-pixel convolutions at the keypoint descriptor head.

### 3.1 KEYPOINTNET: NEURAL KEYPOINT DETECTOR AND DESCRIPTOR LEARNING

**Detector Learning.** Following Christiansen et al. (2019), the keypoint head outputs a location relative to the $8 \times 8$ grid in which it operates for each pixel in the encoder embedding: $[u'_i, v'_i]$. The corresponding input image resolution coordinates $[u_i, v_i]$ are computed taking into account the grid's position in the encoder embedding. We compute the corresponding keypoint location $[u^*_i, v^*_i]$ in the target frame after warping via the known homography **H**. For each warped keypoint, the closest corresponding keypoint in the target frame is associated based on Euclidean distance. We discard keypoint pairs for which the distance is larger than a threshold $\epsilon_{uv}$. The associated keypoints in the target frame are denoted by $\hat{\mathbf{p}}_t = \{[\hat{u}_t, \hat{v}_t]\}$. We optimize keypoint locations using the following self-supervised loss formulation, which enforces keypoint location consistency across different views of the same scene:

$$L_{loc} = \sum_i ||\mathbf{p}^*_t - \hat{\mathbf{p}}_t||_2 \ .$$

(1)

As described earlier, the method of Christiansen et al. (2019) does not allow the predicted keypoint locations for each cell to cross cell-boundaries. Instead, we propose a novel formulation which allows us to effectively aggregate keypoints across cell boundaries. Specifically, we map the relative cell coordinates $[u'_s, v'_s]$ to input image coordinates via the following function:

$$[v_i, u_i] = [row_i^{center}, col_i^{center}] + [v'_i, u'_i] \frac{\sigma_1 (\sigma_2 - 1)}{2} \ ,$$
$$v'_i, u'_i \in (-1, 1)$$

(2)

with $\sigma_2 = 8$, i.e. the cell size, and $\sigma_1$ is a ratio relative to the cell size. $row_i^{center}, col_i^{center}$ are the center coordinates of each cell. By setting $\sigma_1$ larger than 1, we allow the network to predict keypoint locations across cell borders. Our formulation predicts keypoint locations with respect to the cell center, and allows the predicted keypoints to drift across cell boundaries. We illustrate this in Figure 3, where we allow the network to predict keypoints outside the cell-boundary, thereby allowing the keypoints especially at the cell-boundaries to be matched effectively. In the ablation study (Section 4.3), we quantify the effect of this contribution and show that it significantly improves the performance of our keypoint detector.

**Descriptor Learning.** As recently shown by Pillai et al. (2019) and Guizilini et al. (2019), subpixel convolutions via pixel-shuffle operations (Shi et al., 2016) can greatly improve the quality of dense predictions, especially in the self-supervised regime. In this work, we include a fast upsampling step

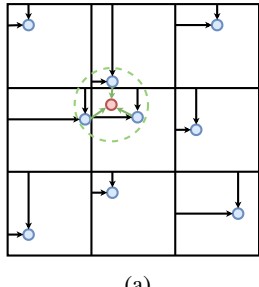 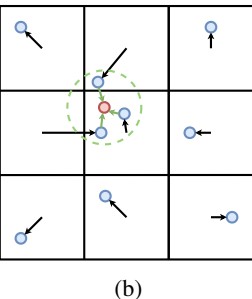

(a)            (b)

Figure 3: Cross-border detection illustration. We illustrate how a warped point (in red) can be associated to multiple predicted points (in blue) based on a distance threshold (dashed circle). (a) The keypoint location head in Christiansen et al. (2019) forces keypoint predictions in the same cell, causing convergence issues since these points can only be pulled up to the cell-boundary. (b) Instead, in our method, we design the network to predict the localization from the cell-center and allow keypoints to be outside the border for better matching and aggregation.

before regressing the descriptor, which promotes the capture of finer details in a higher resolution grid. The architectural diagram of the descriptor head is show in Figure 2. In the ablative analysis (Section 4.3), we show that the addition of this step greatly improves the quality of our descriptors.

We employ metric learning for training the descriptors. While the contrastive loss (Hadsell et al., 2006) is commonly used in the literature for this task, we propose to use a per-pixel triplet loss (Schroff et al., 2015) with nested hardest sample mining as described in Tang et al. (2018) to train the descriptor. Recall that each keypoint $p_i \in \mathbf{p_s}$ in the source image has associated descriptor $f_i$, an *anchor* descriptor, which we obtain by sampling the appropriate location in the dense descriptor map $\mathbf{f_s}$ as described in DeTone et al. (2018b). The associated descriptor $f_{i,+}^*$ in the target frame, a *positive* descriptor, is obtained by sampling the appropriate location in the target descriptor map $\mathbf{f_t}$ based on the warped keypoint position $p_i^*$. The nested triplet loss is therefore defined as:

$$L_{desc} = \sum_i \max(0, \|\mathbf{f}_i, \mathbf{f}_{i,+}^*\|_2 - \|\mathbf{f}_i, \mathbf{f}_{i,-}^*\|_2 + m) \,, \tag{3}$$

which minimizes the distance between the anchor and positive descriptors, and maximizes the distance between the anchor and a *negative* $\mathbf{f}_{i,-}^*$ sample. We pick the negative sample which is the closest in the descriptor space that is not a positive sample. Any sample other than the true match can be used as the negative pair for the anchor, but the hardest negative sample will contribute the most to the loss function, and thereby accelerating the metric learning. Here $m$ denotes the distance margin enforcing how far dissimilar descriptors should be pushed away in the descriptor space.

**Score Learning.** The third head of the decoder is responsible for outputting the score associated with each descriptor. At test time, this value will indicate the most reliable keypoints from which a subset will be selected. Thus the objective of $L_{score}$ is two-fold: (i) we want to ensure that feature-pairs have consistent scores, and (ii) the network should learn that good keypoints are the ones with low feature point distance. Following Christiansen et al. (2019) we achieve this objective by minimizing the squared distance between scores for each keypoint-pair, and minimizing or maximizing the average score of a keypoint-pair if the distance between the paired keypoints is greater or less than the average distance respectively:

$$L_{score} = \sum_i \left[ \frac{(s_i + \hat{s}_i)}{2} \cdot (d(p_i, \hat{p}_i) - \bar{d}) + (s_i - \hat{s}_i)^2 \right] \tag{4}$$

Here, $s_i$ and $\hat{s}_i$ are the scores of the source and target frames respectively, and $\bar{d}$ is the average reprojection error of associated points in the current frame, $\bar{d} = \sum_i^L \frac{d(p_i, \hat{p}_i)}{L}$, with $d$ being the feature distance in 2D Euclidean space and $L$ being the total number of feature pairs.

## 3.2 IO-Net: Neural Outlier Rejection as an Auxiliary Task

Keypoint and descriptor learning is a task which is tightly coupled with outlier rejection. In this work, we propose to use the latter as a proxy task to supervise the former. Specifically, we associate keypoints from the source and target images based on *descriptor distance*: $\{\mathbf{p_s}, \mathbf{p_t^*}, x(\mathbf{f_s}, \mathbf{f_t^*})\}$. In addition, only keypoints with the lowest K predicted scores are used for training. Similar to the hardest sample mining, this approach accelerates the converging rate and encourages the generation of a richer supervisory signal from the outlier rejection loss. To disambiguate the earlier association of keypoint pairs based on reprojected distance defined in Section 3.1, we denote the distance metric by $x$ and specify that we refer to Euclidean distance in descriptor space. The resulting keypoint pairs along with the computed distance are passed through our proposed IO-Net which outputs the probability that each pair is an inlier or outlier. Formally, we define the loss at this step as:

$$L_{IO} = \sum_i \frac{1}{2}(r_i - \mathbf{sign}(\|p_i^* - \hat{p}_i\|_2 - \epsilon_{uv}))^2 \, , \tag{5}$$

where $r$ is the output of the IO-Net, while $\epsilon_{uv}$ is the same Euclidean distance threshold used in Section 3. Different from a normal classifier, we also back propagate the gradients back to the input sample, i.e., $\{\mathbf{p_s}, \mathbf{p_t^*}, x(\mathbf{f_s}, \mathbf{f_t^*})\}$, thus allowing us to optimize both the location and descriptor for these associated point-pairs in an end-to-end differentiable manner.

The outlier rejection task is related to the neural network based RANSAC (Brachmann & Rother, 2019) in terms of the final goal. In our case, since the ground truth homography transform $H$ is known, the random sampling and consensus steps are not required. Intuitively, this can be seen as a special case where only one hypothesis is sampled, i.e. the ground truth. Therefore, the task is simplified to directly classifying the outliers from the input point-pairs. Moreover, a second difference with respect to existing neural RANSAC methods arises from the way the outlier network is used. Specifically, we use the outlier network to explicitly generate an additional proxy supervisory signal for the input point-pairs, as opposed to rejecting outliers.

The final training objective we optimize is defined as:

$$\mathcal{L} = \alpha L_{loc} + \beta L_{desc} + \lambda L_{score} + L_{IO} \, , \tag{6}$$

where $[\alpha, \beta, \lambda]$ are weights balancing different losses.

## 4 Experimental Results

### 4.1 Datasets

We train our method using the COCO dataset (Lin et al., 2014), specifically the 2017 version which contains $118k$ training images. Note that we solely use the images, without any training labels, as our method is completely self-supervised. Training on COCO allows us to compare against SuperPoint (DeTone et al., 2018b) and UnsuperPoint (Christiansen et al., 2019), which use the same data for training. We evaluate our method on image sequences from the HPatches dataset (Balntas et al., 2017), which contains 57 illumination and 59 viewpoint sequences. Each sequence consists of a reference image and 5 target images with varying photometric and geometric changes for a total of 580 image pairs. In Table 2 and Table 3 we report results averaged over the whole dataset. And for fair comparison, we evaluate results generated without applying Non-Maxima Suppression (NMS).

To evaluate our method and compare with the state-of-the-art, we follow the same procedure as described in (DeTone et al., 2018b; Christiansen et al., 2019) and report the following metrics: Repeatability, Localization Error, Matching Score (M.Score) and Homography Accuracy. For the Homography accuracy we use thresholds of 1, 3 and 5 pixels respectively (denoted as Cor-1, Cor-3 and Cor-5 in Table 3). The details of the definition of these metrics can be found in the appendix.

### 4.2 Implementation details

We implement our networks in PyTorch (Paszke et al., 2017) and we use the ADAM (Kingma & Ba, 2014) optimizer. We set the learning rate to $10^{-3}$ and train for 50 epochs with a batch size of 8, halving the learning rate once after 40 epochs of training. The weights of both networks are

| Method | Repeat. ↑ | Loc. ↓ | Cor-1 ↑ | Cor-3 ↑ | Cor-5 ↑ | M.Score ↑ |
|---|---|---|---|---|---|---|
| V0 - Baseline | 0.633 | 1.044 | 0.503 | 0.796 | 0.868 | 0.491 |
| V1 - Cross | **0.689** | 0.935 | 0.491 | 0.805 | 0.874 | 0.537 |
| V2 - CrossUpsampling | 0.686 | 0.918 | 0.579 | 0.866 | 0.916 | **0.544** |
| V3 - IO-Net | 0.685 | **0.885** | 0.602 | 0.836 | 0.886 | 0.520 |
| V4 - Proposed | 0.686 | 0.890 | 0.591 | **0.867** | 0.912 | **0.544** |

Table 1: Ablative comparison for 5 different configurations where V0: Baseline, V1: V0 + Cross-border detection, V2: V1 + Descriptor up-sampling, V3: V2 + $L_{IO}$, and finally the proposed method V4: V3 + $L_{desc}$. In general, the results indicate that for most metrics, the proposed method is within reasonable margin of the best-performing model variant, while achieving strong generalization performance across all performance metrics including repeatability, localization error, homography accuracy and matching score.

randomly initialized. We set the weights for the total training loss as defined Equation (6) to $\alpha = 1$, $\beta = 2$, and $\lambda = 1$. These weights are selected to balance the scales of different terms. We set $\sigma_1 = 2$ in order to avoid border effects while maintaining distributed keypoints over image, as described in Section 3.1. The triplet loss margin $m$ is set to $0.2$. The relaxation criteria $c$ for negative sample mining is set to $8$. When training the outlier rejection network described in Section 3.2, we set $K = 300$, i.e. we choose the lowest 300 scoring pairs to train on.

We perform the same types of homography adaptation operations as DeTone et al. (2018b): crop, translation, scale, rotation, and symmetric perspective transform. After cropping the image with $0.7$ (relative to the original image resolution), the amplitudes for other transforms are sampled uniformly from a pre-defined range: scale $[0.8, 1.2]$, rotation $[0, \frac{\pi}{4}]$ and perspective $[0, 0.2]$. Following Christiansen et al. (2019), we then apply non-spatial augmentation separately on the source and target frames to allow the network to learn illumination invariance. We add random per-pixel Gaussian noise with magnitude $0.02$ (for image intensity normalized to $[0, 1]$) and Gaussian blur with kernel sizes $[1, 3, 5]$ together with color augmentation in brightness $[0.5, 1.5]$, contrast $[0.5, 1.5]$, saturation $[0.8, 1.2]$ and hue $[-0.2, 0.2]$. In addition, we randomly shuffle the color channels and convert color image to gray with probability $0.5$.

### 4.3 ABLATIVE STUDY

In this section, we evaluate five different variants of our method. All experiments described in this section are performed on images of resolution 240x320. We first define V0-V2 as (i) V0: baseline version with cross border detection and descriptor upsampling disabled; (ii) V1: V0 with cross border detection enabled; (iii) V2: V1 with descriptor upsampling enabled. These three variants are trained *without* neural outlier rejection, while the other two variants are (iv) V3: V2 with descriptor trained using *only* $L_{IO}$ and *without* $L_{desc}$ and finally (v) V4 - proposed: V3 together with $L_{desc}$ loss. The evaluation of these methods is shown in Table 1. We notice that by avoiding the border effect described in Section 3.1, *V1* achieves an obvious improvement in Repeatablity as well as the Matching Score. Adding the descriptor upsampling step improves the matching performance greatly without degrading the Repeatability, as can be seen by the numbers reported under *V2*. Importantly, even though *V3* is trained without the descriptor loss $L_{desc}$ defined in Section 3.1, we note further improvements in matching performance. This validates our hypothesis that the proxy task of inlier-outlier prediction can generate supervision for the original task of keypoint and descriptor learning. Finally, by adding the triplet loss, our model reported under *V4 - Proposed* achieves good performance which is within error-margin of the best-performing model variant, while achieving strong generalization performance across all performance metrics including repeatability, localization error, homography accuracy and matching score.

To quantify our runtime performance, we evaluated our model on a desktop with an Nvidia Titan Xp GPU on images of 240x320 resolution. We recorded $174.5$ FPS and $194.9$ FPS when running our model with and without the descriptor upsampling step.

### 4.4 PERFORMANCE EVALUATION

In this section, we compare the performance of our method with the state-of-the-art, as well as with traditional methods on images of resolutions $240 \times 320$ and $480 \times 640$ respectively. For the results obtained using traditional features as well as for LF-Net (Ono et al., 2018) and SuperPoint (DeTone

| Method | Repeat. ↑ | | Loc. Error ↓ | |
|---|---|---|---|---|
| | 240x320 | 480x640 | 240x320 | 480x640 |
| ORB | 0.532 | 0.525 | 1.429 | 1.430 |
| SURF | 0.491 | 0.468 | 1.150 | 1.244 |
| BRISK | 0.566 | 0.505 | 1.077 | 1.207 |
| SIFT | 0.451 | 0.421 | 0.855 | 1.011 |
| LF-Net(indoor) (Ono et al., 2018) | 0.486 | 0.467 | 1.341 | 1.385 |
| LF-Net(outdoor) (Ono et al., 2018) | 0.538 | 0.523 | 1.084 | 1.183 |
| SuperPoint (DeTone et al., 2018b) | 0.631 | 0.593 | 1.109 | 1.212 |
| UnsuperPoint (Christiansen et al., 2019) | 0.645 | 0.612 | **0.832** | 0.991 |
| Proposed | **0.686** | **0.684** | 0.890 | **0.970** |

Table 2: Our proposed method outperforms all the listed traditional and learned feature detectors in repeatability (higher is better). For localization error (lower is better), UnsuperPoint performs better in lower resolution images while our method performs better for higher resolutions.

| Method | 240x320, 300 points | | | | 480 x 640, 1000 points | | | |
|---|---|---|---|---|---|---|---|---|
| | Cor-1 | Cor-3 | Cor-5 | M.Score | Cor-1 | Cor-3 | Cor-5 | M.Score |
| ORB | 0.131 | 0.422 | 0.540 | 0.218 | 0.286 | 0.607 | 0.71 | 0.204 |
| SURF | 0.397 | 0.702 | 0.762 | 0.255 | 0.421 | 0.745 | 0.812 | 0.230 |
| BRISK | 0.414 | 0.767 | 0.826 | 0.258 | 0.300 | 0.653 | 0.746 | 0.211 |
| SIFT | **0.622** | 0.845 | 0.878 | 0.304 | **0.602** | 0.833 | 0.876 | 0.265 |
| LF-Net(indoor) | 0.183 | 0.628 | 0.779 | 0.326 | 0.231 | 0.679 | 0.803 | 0.287 |
| LF-Net(outdoor) | 0.347 | 0.728 | 0.831 | 0.296 | 0.400 | 0.745 | 0.834 | 0.241 |
| SuperPoint | 0.491 | 0.833 | 0.893 | 0.318 | 0.509 | 0.834 | 0.900 | 0.281 |
| UnsuperPoint | 0.579 | 0.855 | 0.903 | 0.424 | 0.493 | 0.843 | 0.905 | 0.383 |
| Proposed | 0.591 | **0.867** | **0.912** | **0.544** | 0.564 | **0.851** | **0.907** | **0.510** |

Table 3: Homography estimation accuracy with 3 different pixel distance thresholds (i.e. Cor-1,3 and 5) and matching performance comparison. As shown, our proposed method outperforms all the listed traditional and learning based methods except the one case where SIFT performs the best.

et al., 2018b) we report the same numbers as computed by (Christiansen et al., 2019). During testing, keypoints are extracted in each view keeping the top 300 points for the lower resolution and 1000 points for the higher resolution from the score map. The evaluation of keypoint detection is shown in Table 2. For *Repeatibility*, our method notably outperforms other methods and is not significantly affected when evaluated with different image resolutions. For the *Localization Error*, UnsuperPoint performs better in lower resolution image while our method performs better for higher resolution.

The homography estimation and matching performance results are shown in Table 3. In general, self-supervised learning methods provide keypoints with higher matching score and better homography estimation for the *Cor-3* and *Cor-5* metrics, as compared to traditional handcrafted features (e.g. SIFT). For the more stringent threshold *Cor-1*, SIFT performs the best, however, our method outperforms all other learning based methods. As shown in Table 1, our best performing model for this metric is trained using only supervision from the outlier rejection network, without the triplet loss. This indicates that, even though fully self-supervised, this auxiliary task can generate high quality supervisory signals for descriptor training. We show additional qualitative and qualitative results of our method in the appendix.

## 5   CONCLUSION

In this paper, we proposed a new learning scheme for training a keypoint detector and associated descriptor in a self-supervised fashion. Different with existing methods, we used a proxy network to generate an extra supervisory signal from a task tightly connected to keypoint extraction: outlier rejection. We show that even without an explicit keypoint descriptor loss in the IO-Net, the supervisory signal from the auxiliary task can be effectively propagated back to the keypoint network to generate distinguishable descriptors. Using the combination of the proposed method as well as the improved network structure, we achieve competitive results in the homography estimation benchmark.

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

## A   HOMOGRAPHY ESTIMATION EVALUATION METRIC

We evaluated our results using the same metrics as DeTone et al. (2018b). The Repeatability, Localization Error and Matching Score are generated with a correctness distance threshold of 3. All the metrics are evaluated from both view points for each image pair.

**Repeatability.** The repeatability is the ratio of correctly associated points after warping into the target frame. The association is performed by selecting the closest in-view point and comparing the distance with the correctness distance threshold.

**Localization Error.** The localization error is calculated by averaging the distance between warped points and their associated points.

**Matching Score (M.Score).** The matching score is the success rate of retrieving correctly associated points through nearest neighbour matching using descriptors.

**Homography Accuracy.** The homography accuracy is the success rate of correctly estimating the homographies. The mean distance between four corners of the image planes and the warped image planes using the estimated and the groundtruth homography matrices are compared with distances $1, 3, 5$. To estimate the homography, we perform reciprocal descriptor matching, and we use openCV's $findHomography$ method with RANSAC, maximum number of $5000$ iterations, confidence threshold of $0.9995$ and error threshold $3$.

## B   DETAILED QUALITATIVE AND QUANTITATIVE ANALYSIS ON HPATCHES

To capture the variance induced by the RANSAC component during evaluation we perform additional experiments summarized in Table 4 where each entry reports the mean and standard deviation across 10 runs with varying RANSAC seeds. We notice better homography performance on the illumination subset than on the viewpoint subset. This is to be expected as the viewpoint subset contains image pairs with extreme rotation which are problematic for our method which is fully convolutional.

| HPatches subset | Repeat. ↑ | Loc. ↓ | Cor-1 ↑ | Cor-3 ↑ | Cor-5 ↑ | M.Score ↑ |
|---|---|---|---|---|---|---|
| All | 0.686 | 0.890 | 0.595±0.012 | 0.871±0.005 | 0.912±0.005 | 0.544 |
| Illumination | 0.678 | 0.826 | 0.753±0.014 | 0.972±0.004 | 0.984±0.004 | 0.614 |
| Viewpoint | 0.693 | 0.953 | 0.494±0.015 | 0.801±0.008 | 0.857±0.008 | 0.479 |

Table 4: Detailed analysis of the performance of our method: we evaluate on different splits of the HPatches dataset (Illumination, Viewpoint and complete dataset) on images of resolution 320x240. To capture the variance in the evaluation of the homography (Cor-1,3 and 5) due to RANSAC, we perform 10 evaluation runs with different random seeds and report the mean and standard deviation.

We also evaluate our method as well as SIFT and ORB on the *graffiti*, *bark* and *boat* sequences of the HPatches dataset and summarize our results in Table 5, again reporting averaged results over 10 runs. We note that our method consistently outperforms ORB. Our method performs worse than SIFT (which is more robust to extreme rotations) on the bark and boat sequences, but we obtain better results on the graffiti sequence.

Figure 4 denotes examples of successful matching under strong illumination, rotation and perspective transformation. Additionally, we also show our matches on pairs of images from the challenging graffiti, bark and boat sequences of HPatches in Figures 5, 6, and 7. Specifically, the top row in each figure shows our results, while the bottom row shows SIFT. The left sub-figure on each row shows images (1,2) of each sequence, while the right sub-figure shows images (1,6). We note that on images (1,2) our results are comparable to SIFT, while on images (1,6) we get fewer matches. Despite the extreme perspective change, we report that our method is able to successfully match features on images (1,6) of the boat sequence.

| | HPatches subset | Repeat. ↑ | Loc. ↓ | Cor-1 ↑ | Cor-3 ↑ | Cor-5 ↑ | M.Score ↑ |
|---|---|---|---|---|---|---|---|
| **Ours** | Bark | 0.419 | 1.734 | 0.200±0.000 | 0.200±0.000 | 0.200±0.000 | 0.088 |
| | Boat | 0.588 | 1.357 | 0.260±0.092 | 0.380±0.060 | 0.400±0.000 | 0.223 |
| | Graffiti | 0.670 | 1.039 | 0.340±0.092 | 0.720±0.098 | 0.780±0.060 | 0.336 |
| **ORB** | Bark | 0.620 | 1.259 | 0.000±0.000 | 0.060±0.092 | 0.180±0.060 | 0.110 |
| | Boat | 0.812 | 1.127 | 0.000±0.000 | 0.420±0.108 | 0.560±0.120 | 0.226 |
| | Graffiti | 0.722 | 1.164 | 0.080±0.098 | 0.340±0.092 | 0.400±0.000 | 0.210 |
| **SIFT** | Bark | 0.470 | 1.252 | 0.600±0.000 | 0.960±0.080 | 0.960±0.080 | 0.292 |
| | Boat | 0.527 | 0.928 | 0.560±0.080 | 0.980±0.060 | 1.000±0.000 | 0.351 |
| | Graffiti | 0.515 | 1.277 | 0.340±0.092 | 0.560±0.120 | 0.660±0.092 | 0.237 |

Table 5: Detailed analysis of the performance of our method on the *Bark*, *Boat* and *Graffitti* sequences of the HPatches dataset. To capture the variance in the evaluation of the homography (Cor-1,3 and 5) due to RANSAC, we perform 10 evaluation runs with different random seeds and report the mean and standard deviation.

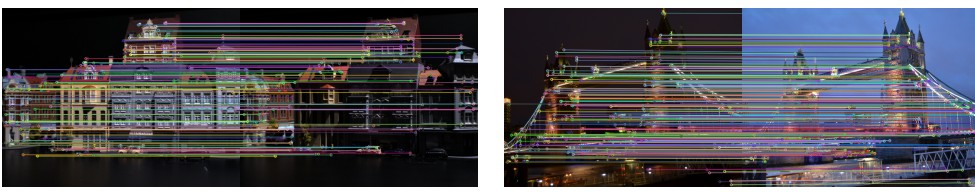

(a) Qualitative results of our method for illumination cases.

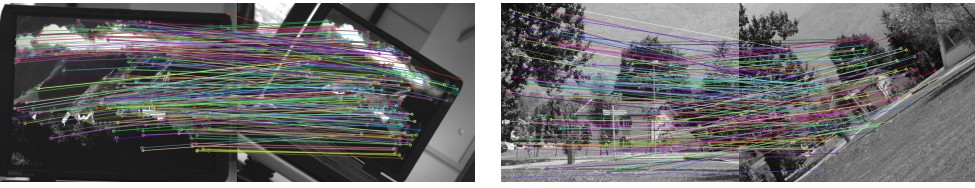

(b) Qualitative results of our method for rotation cases.

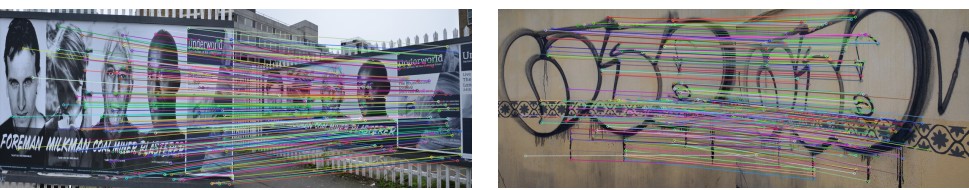

(c) Qualitative results of our method for perspective cases.

Figure 4: Qualitative results of our method on images pairs of the HPatches dataset.

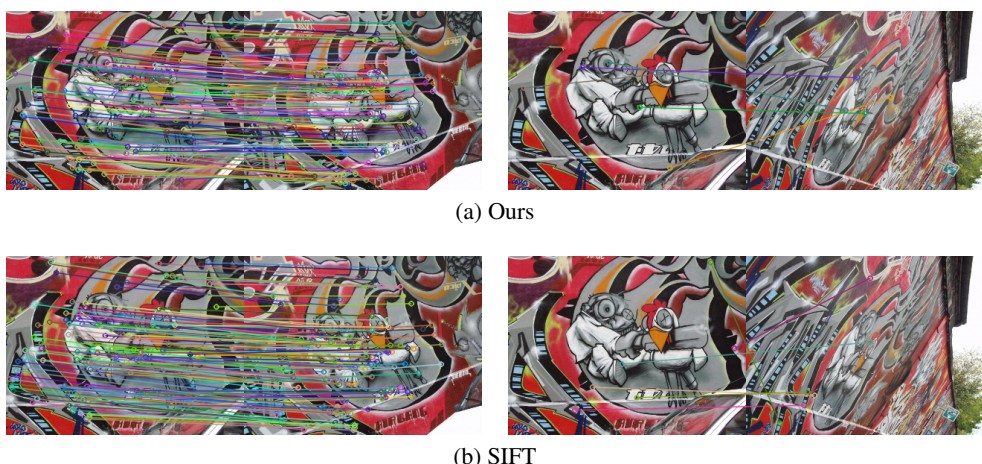

Figure 5: Qualitative results of our method vs SIFT on the "graffiti" subset of images of the HPatches dataset.

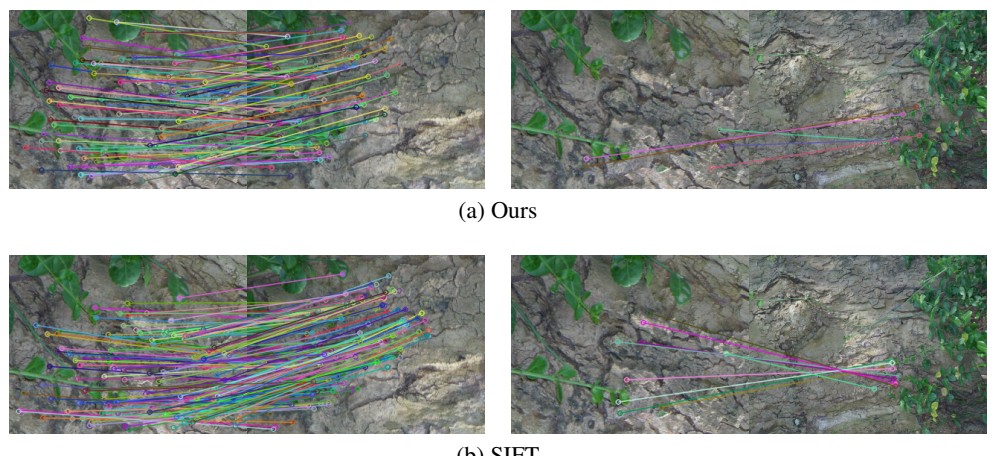

Figure 6: Qualitative results of our method vs SIFT on the "bark" subset of images of the HPatches dataset.

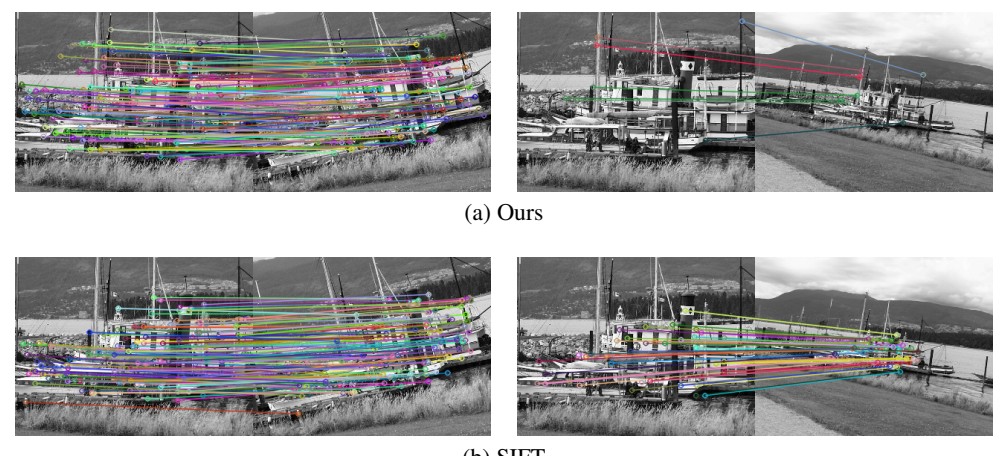

Figure 7: Qualitative results of our method vs SIFT on the "boat" subset of images of the HPatches dataset.

## C  ARCHITECTURE DIAGRAM

|      | Layer Description | K | Output Tensor Dim. |
|------|-------------------|---|--------------------|
| #0   | Input RGB image   |   | 3×H×W              |
| **Encoder** | | | |
| #1   | Conv2d + BatchNorm + LReLU            | 3 | 32×H×W       |
| #2   | Conv2d + BatchNorm + LReLU + Dropout  | 3 | 32×H×W       |
| #3   | Max. Pooling (× 1/2)                  | 3 | 32×H/2×W/2   |
| #4   | Conv2d + BatchNorm + LReLU            | 3 | 64×H/2×W/2   |
| #5   | Conv2d + BatchNorm + LReLU + Dropout  | 3 | 64×H/2×W/2   |
| #6   | Max. Pooling (× 1/2)                  | 3 | 64×H/4×W/4   |
| #7   | Conv2d + BatchNorm + LReLU            | 3 | 128×H/4×W/4  |
| #8   | Conv2d + BatchNorm + LReLU + Dropout  | 3 | 128×H/4×W/4  |
| #9   | Max. Pooling (× 1/2)                  | 3 | 128×H/8×W/8  |
| #10  | Conv2d + BatchNorm + LReLU            | 3 | 256×H/8×W/8  |
| #11  | Conv2d + BatchNorm + LReLU + Dropout  | 3 | 256×H/8×W/8  |
| **Score Head** | | | |
| #12  | Conv2d + BatchNorm + Dropout (#11)    | 3 | 256×H/8×W/8  |
| #13  | Conv2d + Sigmoid                      | 3 | 1×H/8×W/8    |
| **Location Head** | | | |
| #14  | Conv2d + BatchNorm + Dropout (#11)    | 3 | 256×H/8×W/8  |
| #15  | Conv2d + Tan. Hyperbolic              | 3 | 2×H/8×W/8    |
| **Descriptor Head** | | | |
| #16  | Conv2d + BatchNorm + Dropout (#11)    | 3 | 256×H/8×W/8  |
| #17  | Conv2d + BatchNorm                    | 3 | 512×H/8×W/8  |
| #18  | Pixel Shuffle (× 2)                   | 3 | 128×H/8×W/8  |
| #19  | Conv2d + BatchNorm (#8 ⊗ #18)         | 3 | 256×H/4×W/4  |
| #20  | Conv2d                                | 3 | 256×H/4×W/4  |

Table 6: KeyPointNet diagram, composed of an encoder followed by three decoder heads. The network receives as input an RGB image and returns scores, locations and descriptors. Numbers in parenthesis indicate input layers, ⊗ denotes feature concatenation, and we used 0.2 dropout values.

|      | Layer Description | K | Output Tensor Dim. |
|------|-------------------|---|--------------------|
| #0   | Input KeyPoints   |   | 5×N                |
| #1   | Conv1d + ReLU           | 1 | 128×N |
| #2   | ResidualBlock           | - | 128×N |
| #3   | ResidualBlock (#2 ⊕ #1) | - | 128×N |
| #4   | ResidualBlock (#3 ⊕ #2) | - | 128×N |
| #5   | ResidualBlock (#4 ⊕ #3) | - | 128×N |
| #6   | Conv1d                  | 1 | 1×N   |
| **ResidualBlock** | | | |
|      | Conv1d + InstNorm + BatchNorm + ReLU | 1 | 128×N |
|      | Conv1d + InstNorm + BatchNorm + ReLU | 1 | 128×N |

Table 7: IO-Net diagram, composed of 4 residual blocks. The network receives as input a series of 5-dimensional vector consists of keypoint pair and descriptor distance, outputs a binary inlier-outlier classification. Numbers in parenthesis indicate input layers, and ⊕ denotes feature addition.

