# OpenReview forum: "Neural Outlier Rejection for Self-Supervised Keypoint Learning"
_ICLR.cc/2020/Conference — Accept (Poster)_

### Official Review · AnonReviewer3 · 2019-10-16
**Official Blind Review #3**

**Rating:** 8

**Review:**

The paper is devoted to self-supervised learning of local features (both detectors and descriptors simultaneously). The problem is old yet not fully solved yet, because handcrafted SIFT is still winning the benchmarks. This work mostly follows and improves upon SuperPoint (DeTone et.al 2017) and the follow-up work UnsuperPoint (Christiansen et.al 2019) architecture and training scheme.

The claimed contributions are following:
  - use the recently published Neural Guided RANSAC as additional auxilary loss provider
  - allowing the "cells" to predict keypoint location outside the cell while learning
  - special procedure for improving descriptors interpolation


The experiments are performed on HSequences dataset (wrongly called "HPatches", as HPatches dataset is literally image patches, not full images), showing noticable improvement over the state of the art.


Strong points:

 - Method is sound, paper is mostly well written and results are good (may be too good, see questions).



 Questions:

  1) Regarding descriptor interpolation, which is claimed as contribution. It is not clear for me, how different it is compared to SuperPoint one, which also do descriptor upsampling, so that network output is H x W x [256], i.e. full resolution. Could you please clarify the differences to it? Also, in Figure 2 it is not clear, how one can do "feature concatenation" for blocks with different spatial resolution.

  2) Why the IONet is used only for training?  Wouldn`t it better to actually learn everything end-to-end, which is already done in paper and evaluate?

  3) How is association in training (e.g. on Fig.3) done, if multiple cells in img2 returns keypoint close to the same keypoint in img1?

  4) HSequences consists of two subsets: Illumination and Viewpoint. Could you please report results per subset instead of per whole dataset? Could you also please specifically report results for the following image sequences: graffity, bark, boat, especially for 1-6 pairs and visualize matches (same way as in Figure 4-6)?
  The reason that I am asking these, is results looks like too well and I suspect overfitting to a points, which are suitable for estimation (small) homography, not general-purpose points.

  5) Could you please explain in more details, how did you do homography estimation precision benchmark? Specifically, was Lowe`s second nearest neighbor ratio used for filtering out wrong matches? If not, could you please repeat this experiments with it, at least for SIFT matches?


Small comments:
    - list of contributions in abstract is inconsistent with 3rd paragraph in Introduction, which also lists contributions.


***
Overall I like the work, but there are unclear moments to me.


****
After rebuttal comments. While this paper may appear not "sexy" I think it is quite valuable for the local features learning community: both for the main contributions, and small details and tricks evaluated inside.
I am happy to increase my score to strong accept.

**Experience Assessment:**

I have published in this field for several years.

**Review Assessment: Checking Correctness Of Derivations And Theory:**

N/A

**Review Assessment: Checking Correctness Of Experiments:**

I carefully checked the experiments.

**Review Assessment: Thoroughness In Paper Reading:**

I read the paper thoroughly.

---

> ### Author Response · Authors · 2019-11-14
> **Response to Reviewer #3 (part 2/2)**
>
> R3: How is association in training (e.g. on Fig.3) done, if multiple cells in img2 returns keypoint close to the same keypoint in img1?
>
> We have updated the caption in Figure 3 to properly explain the differences between each scenario. In particular, the UnsuperPoint method in (a) forces keypoint predictions to be in the same cell. Our method in (b) predicts locations from the cell-center and allow keypoints to  cross cell borders, which promotes better matching and aggregation. This implies that multiple keypoints from one image (e.g. blue keypoints) may have the same corresponding keypoint in the second image (e.g. red keypoints) which is the desirable behavior we aim for.
>
> R3: HSequences consists of two subsets: Illumination and Viewpoint. Could you please report results per subset instead of per whole dataset? Could you also please specifically report results for the following image sequences: graffiti, bark, boat, especially for 1-6 pairs and visualize matches (same way as in Figure 4-6)?
>
> We thank the reviewer for this comment - it allows us to further explore the performance of our method. We added Table 4 in the appendix, where we evaluate the performance of our method on the two subsets of HPatches. We mention that each entry reports the mean and standard deviation across 10 runs with varying RANSAC seeds, to also evaluate any randomness induced during the evaluation. We notice better homography performance on the illumination subset than on the viewpoint subset. This is to be expected as the viewpoint subset contains image pairs with extreme rotation which are problematic for our method which is fully convolutional.
> We evaluate the Graffiti, Bark and Boat sequences of the HPatches dataset, and report the results of our method as well as SIFT and ORB in Table 5 in the appendix, again reporting averaged results over 10 runs. We note that our method consistently outperforms ORB. Our method performs worse than SIFT (which is more robust to extreme rotations) on the bark and boat sequences, but we obtain better results on the graffiti sequence.
> Finally, we added Figures 5, 6 and 7 to qualitatively show our matches on pairs of images from the graffiti, bark and boat sequences. Specifically, the top row in each figure shows our results, while the bottom row shows SIFT. The left subfigure on each row shows images (1,2) of each sequence, while the right subfigure shows images (1,6). We note that on images (1,2) our results are comparable to SIFT, while on images (1,6) we get fewer matches. Despite the extreme perspective change, we report that our method is able to successfully match features on images (1,6) of the boat sequence.
>
> R3: Could you please explain in more details, how did you do homography estimation precision benchmark? Specifically, was Lowe`s second nearest neighbor ratio used for filtering out wrong matches? If not, could you please repeat this experiments with it, at least for SIFT matches?
>
> To estimate the homography, we performed reciprocal descriptor matching, and we only kept matches that have each other as nearest neighbors, e.g. if the nearest neighbor of a descriptor X from image A is descriptor Y in image B, then the nearest neighbor of descriptor Y is descriptor X in image A. We did not use Lowe’s second nearest neighbor ratio which we found to yield worse results. To compute the homography, we used OpenCV’s findHomography method with RANSAC, error threshold 3 and a maximum of 5000 iterations.
> Please see below the results of our evaluation of SIFT on 320x240 and 640x480 on HPatches with and without Lowe’s second nearest neighbor ratio:
>
> 320x240:
> Method                 |    Repeat.  |    Loc.   |  Cor-1  | Cor-3 |  Cor-5 |   M.Score
> SIFT+ 0.7 ratio     |      0.474   |  0.993   |  0.497   | 0.752 | 0.798  |   0.287
> SIFT+ 0.8 ratio     |      0.474   |  0.993   |  0.529   | 0.788 | 0.852  |   0.287
> SIFT+ 0.9 ratio     |      0.474   |  0.993   |  0.564   | 0.836 | 0.879  |   0.287
> SIFT + reciprocal |      0.474    |  0.993   |  0.605  | 0.859 | 0.895  |    0.287
>
> 640x480:
> Method                 |    Repeat.  |    Loc.   |  Cor-1  | Cor-3 |  Cor-5 |   M.Score
> SIFT+ 0.7 ratio     |     0.498    |  1.062   |  0.507   | 0.790 | 0.862  |   0.290
> SIFT+ 0.8 ratio     |     0.498    |  1.062   |  0.548   | 0.826 | 0.905  |   0.290
> SIFT+ 0.9 ratio     |     0.498    |  1.062   |  0.571   | 0.862 | 0.910  |   0.290
> SIFT + reciprocal |      0.498    |  1.062   |  0.584  | 0.864 | 0.917  |    0.290
>
> R3: List of contributions in abstract is inconsistent with 3rd paragraph in Introduction, which also lists contributions.
> We appreciate that the reviewer identified this inconsistency. We have updated the abstract and introduction to clarify the contributions of the paper in a consistent manner.
>
> [1] Shi, Wenzhe, et al. "Real-time single image and video super-resolution using an efficient sub-pixel convolutional neural network." Proceedings of the IEEE CVPR. 2016.

---

> ### Author Response · Authors · 2019-11-14
> **Response to Reviewer #3 (part 1/2)**
>
> R3: The experiments are performed on HSequences dataset (wrongly called "HPatches", as HPatches dataset is literally image patches, not full images), showing noticable improvement over the state of the art.
>
> We thank the reviewer for this clarification. We updated the text to emphasize the fact that we are evaluating on image sequences from the HPatches dataset.
>
> R3: Regarding descriptor interpolation, which is claimed as contribution. It is not clear for me, how different it is compared to SuperPoint one, which also do descriptor upsampling, so that network output is H x W x [256], i.e. full resolution. Could you please clarify the differences to it?
>
> Both our method and SuperPoint perform interpolation to the original image resolution H x W x [256]. The key difference is that our network includes a learneable upsampling step before the interpolation, which goes from H/8 x W/8 x [256] to H/4 x W/4 x [256] based on the work of [1].
>
> R3: Also, in Figure 2 it is not clear, how one can do "feature concatenation" for blocks with different spatial resolution.
>
> We thank the reviewer for pointing out this mistake, we have fixed it and now the feature concatenation process should be correctly illustrated in Figure 2. Diagrams detailing our two networks (KeyPointNet and IO-Net)  were also added to the Appendix.
>
> R3: Why the IONet is used only for training?  Wouldn’t it better to actually learn everything end-to-end, which is already done in paper and evaluate?
>
> The reasons why we only use IO-Net for training are two-fold:
> (1) Our major contribution in this paper is showing that KeyPointNet can learn from a proxy task (IO-Net) to either train the descriptor directly or to improve it (when trained with the descriptor loss). We focus on improving the actual performance of KeyPointNet at training time, rather than using another network to improve the final matching during inference. Moreover, for a fair comparison with other methods we aimed to keep the homography component estimation the same (i.e. using OpenCV and RANSAC), thus showing that our superior results are due to our improved keypoints and descriptors.
> (2) Inspired by Negative Sample mining, which is commonly deployed for metric learning, we perform a similar strategy by feeding the lowest-k score (non-border) keypoints to the IO-Net during training. We found that using the top-k score keypoints performs worse. However, at test time, it is not meaningful to take the keypoints with lowest score to estimate the homography, hence we do not use IO-Net for the evaluation.

---

### Official Review · AnonReviewer1 · 2019-10-23
**Official Blind Review #1**

**Rating:** 6

**Review:**

# UPDATE following rebuttal

Score increased to 6 due to architecture details in supplemental.


# Contributions

The paper contributes a self-supervised method of jointly learning 2D keypoint locations, descriptors, and scores given an input RGB image. The paper builds on previous work, adding:

* A more expressive keypoint location regression, which allows each 8x8 pixel region to vote for a keypoint location outside its boundary

* An upsampling step, similar to a U-net, to allow descriptors to be regressed with more detailed information

* An additional proxy task for the total loss, based on outlier rejection.

The authors train on COCO by manually distorting images to generate pairs with known homography, and show competitive results for keypoint detection and homography estimation tasks.

Decision: Weak reject. I would give this a 5 if the website allowed me to. A more detailed explanation of the neural network architecture, along with minor fixes described below, would make me increase my rating.


I feel the additions to existing pipelines are well motivated but insufficiently explained. In particular, the explanation of the neural network architecture along with figures 1 and 2 leaves many details unclear to me. Phrases like "a 1D CNN ... with 4 default setting residual blocks" is to me insufficient - residual networks have many details such as Resnet V1 or V2 style (ie is there a path right through the network which doesn't hit any activation functions), what kind of normalization is applied, number of channels in each block, how to do skips between different spatial resolutions, etc. The upsampling step for the descriptor head, which is claimed as a novel contribution, is not fully explained - "fast upsampling" implies (correctly) there are many variants of upsampling with different tradeoffs, but from the text I am unsure whether this is nearest neighbour upsampling, a ConvTranspose, etc. Similarly, "VGG style block" leaves some details unclear - whether the resolution downsampling is with a strided convolution / pooling / etc. Lots of the details are implied to be in other previous papers, but I feel that the paper would be hugely improved by exact architectural details.

There are various minor notational discrepancies in the paper - for example the outlier rejection is various defined as "InlierOuterNet (IONet)" and "The Inlier-Outlier model \emph{IO}", which also seems to be the same as the function $C$ defined a paragraph above. Perhaps it is common in this part of the literature, but to me an encoder decoder network is more likely to either be an autoencoder, or for the decoder to output something in the same modality (eg in machine translation). To say that some VGG blocks are an encoder, and the heads which produce keypoint locations / score / descriptor is a decoder, implies all neural networks could be described as an encoder/decoder.

The two figures showing the architecture are very different in design, which is not in itself a problem but the relationship between them could be clearer. I feel that the 'matching' box in figure 1 is misleading because it implies that matching only happens for the IONet, but the loss function for location described in Eq 1 also requires matching keypoints between the image pair. I'm also unclear on the division between direct and indirect supervisory signal - all the 4 loss components have a clear purpose, but it's not obvious what this partitioning means. "Indirect" only appears in this figure and the caption - perhaps.

The term "Anchor" appears only once with no reference, below equation 3 - I appreciate this is an existing term in this subfield, but given that the start of section 3 goes as far as explicitly defining what it means to produce 2D keypoints for an image, I feel defining this term would make the descriptor loss much clearer.

One of the main contributions, that of allowing locations to regress outside their 8x8 area, sounds like a good idea but I feel that Figure 3 does not adequately show the benefit. In both a) and b), the blue estimates appear to be roughly as good as each other - clearly from the ablation a large benefit is gained from this innovation but perhaps a better illustrative example could be made here?

On a more positive note, I feel the components of the loss function are in general very clearly motivated and defined, and the description of training & data augmentation hyperparameters appears complete. If the description of the architecture could be improved that would result in a paper very amenable to reproduction.

The experiments are well explained, and the ablation of the various proposed  components is good. I feel table 1 would be improved with error bars - given that the bold best score is not exclusively next to V4, but in many cases the difference between V4 and the best is ~1%, error bars from different training runs might make clearer that V4 is overall the best configuration.

In the conclusion - "even without an explicit loss" - what is the difference between the loss functions used in this work, and an explicit loss?


Minor corrections:

The euclidean distance between descriptors is various notated as $d$ (section 3), $x$ (above equation 5) and $d$ again (below equation 5).

Typos: "normalzation" -> "normalization", "funcion" -> "function", "tripled" -> "triplet".



**Experience Assessment:**

I do not know much about this area.

**Review Assessment: Checking Correctness Of Derivations And Theory:**

I assessed the sensibility of the derivations and theory.

**Review Assessment: Checking Correctness Of Experiments:**

I assessed the sensibility of the experiments.

**Review Assessment: Thoroughness In Paper Reading:**

I read the paper at least twice and used my best judgement in assessing the paper.

---

> ### Author Response · Authors · 2019-11-14
> **Response to Reviewer #1  (part 2/2)**
>
>
> R1: The term "Anchor" appears only once with no reference, below equation 3 - I appreciate this is an existing term in this subfield, but given that the start of section 3 goes as far as explicitly defining what it means to produce 2D keypoints for an image, I feel defining this term would make the descriptor loss much clearer.
> We agree that this term should be explained, and have added in Section 3.3 a proper description of what an anchor is within the context of the paper.
>
> R1: One of the main contributions, that of allowing locations to regress outside their 8x8 area, sounds like a good idea but I feel that Figure 3 does not adequately show the benefit. In both a) and b), the blue estimates appear to be roughly as good as each other - clearly from the ablation a large benefit is gained from this innovation but perhaps a better illustrative example could be made here?
> We have updated the caption in Figure 3 to properly explain the differences between each scenario. In particular, the UnsuperPoint method in (a) forces keypoint predictions to be in the same cell. While our method predicts localization from the cell-center, so keypoints can be cross-border, which promotes better matching and aggregation. In addition, we have also updated the figure to better illustrate the behavior of both methods.
>
> R1: On a more positive note, I feel the components of the loss function are in general very clearly motivated and defined, and the description of training & data augmentation hyperparameters appears complete. If the description of the architecture could be improved that would result in a paper very amenable to reproduction.
> We thank the reviewer for the positive feedback. We have added in the appendix diagrams with implementation details for our networks, to facilitate reproduction (Section D, Tables 6 and 7). We also plan to open-source our code upon publication.
>
> R1: The experiments are well explained, and the ablation of the various proposed  components is good. I feel table 1 would be improved with error bars - given that the bold best score is not exclusively next to V4, but in many cases the difference between V4 and the best is ~1%, error bars from different training runs might make clearer that V4 is overall the best configuration.
> Unfortunately due to time constraints we were not able to re-train all the models to generate the error bars. However, we have evaluated the best model variant (V4) with different seed configurations to capture the variance induced in the sampling and consensus, and report the results of this experiment in the Appendix in Table 4, where we perform 10 evaluation runs with different random seeds and report the mean and standard deviation.
>
>
> R1: In the conclusion - "even without an explicit loss" - what is the difference between the loss functions used in this work, and an explicit loss?
> We refer to an explicit loss as one that is defined over each of the 3 target outputs (score, location, descriptor) as in Equations 1, 3, and 4. The inlier-outlier loss (Equation 5) however, does not penalize the KeyPointNet outputs directly but acts as an indirect supervisory signal that is able to generate distinguishable keypoint descriptors during matching.
>
> [1] Daniel DeTone, Tomasz Malisiewicz, and Andrew Rabinovich. Superpoint: Self-supervised interestpoint detection and description. InProceedings of the IEEE Conference on Computer Vision andPattern Recognition Workshops, pp. 224–236, 2018b.

---

> ### Author Response · Authors · 2019-11-14
> **Response to Reviewer #1  (part 1/2)**
>
> R1: I feel the additions to existing pipelines are well motivated but insufficiently explained. In particular, the explanation of the neural network architecture along with figures 1 and 2 leaves many details unclear to me. Phrases like "a 1D CNN ... with 4 default setting residual blocks" is to me insufficient - residual networks have many details such as Resnet V1 or V2 style (ie is there a path right through the network which doesn't hit any activation functions), what kind of normalization is applied, number of channels in each block, how to do skips between different spatial resolutions, etc.
>
> We agree that many details involved in the description and implementation of our networks have been left out of the original paper because of the page limitation. To address this, we have added these details in the Appendix diagrams describing both KeyPointNet and IO-Net with sufficient network architecture description. (Section D, Tables 6 and 7).
>
> R1: There are various minor notational discrepancies in the paper - for example the outlier rejection is various defined as "InlierOuterNet (IONet)" and "The Inlier-Outlier model \emph{IO}", which also seems to be the same as the function  defined a paragraph above. Perhaps it is common in this part of the literature, but to me an encoder decoder network is more likely to either be an autoencoder, or for the decoder to output something in the same modality (eg in machine translation). To say that some VGG blocks are an encoder, and the heads which produce keypoint locations / score / descriptor is a decoder, implies all neural networks could be described as an encoder/decoder.
>
> Thank you for pointing out the notational discrepancy. We have focused on unifying the terminology in the revised paper, and we refer to the outlier rejection network as IO-Net. Regarding the encoder-decoder terminology, we aimed to be consistent with the terminology used in SuperPoint [1], where the detector and descriptor heads are referred to as “decoders” using the same shared “encoded” features as input.
>
> R1: The two figures showing the architecture are very different in design, which is not in itself a problem but the relationship between them could be clearer. I feel that the 'matching' box in figure 1 is misleading because it implies that matching only happens for the IONet, but the loss function for location described in Eq 1 also requires matching keypoints between the image pair. I'm also unclear on the division between direct and indirect supervisory signal - all the 4 loss components have a clear purpose, but it's not obvious what this partitioning means. "Indirect" only appears in this figure and the caption - perhaps.
>
> We thank the reviewer for pointing this out - we have updated the caption of Figures 1 and 2 to better explain the relationship between them as well as the combination of the explicit loss applied directly on the KeyPointNet outputs (score, location and descriptor) and indirect loss derived from IO-Net via the outlier rejection classification task.

---

> > ### Comment · AnonReviewer1 · 2019-11-15
> > **Response to rebuttal**
> >
> > I thank the authors for their detailed response and paper revision. The new version of figure 3 and architecture details in Tables 6 and 7 in particular are good, and address most of my concerns. I am raising my score to a 6.
> >
> > Correction in Table 6 - "Tan Harmonic" -> "Tan Hyperbolic" presumably?

---

> > > ### Author Response · Authors · 2019-11-15
> > > **Response to Reviewer #1**
> > >
> > > We have updated it in the paper. Thank you for pointing this out.

---

### Official Review · AnonReviewer2 · 2019-10-24
**Official Blind Review #2**

**Rating:** 6

**Review:**

The following work proposes several improvements over prior works in unsupervised/self-supervised keypoint-descriptor learning such as Christiansen et al. One improvement is the relaxation of the cell-boundaries for keypoint prediction -- specifically allowing keypoints anchored at the cell's center to be offset into neighboring cells. Another change was the introduction of an inlier-outlier classifier network to be used as a proxy loss for the keypoint position and descriptors. They found the inlier-outlier loss to improve homography accuracy at 1 and 3 pixel thresholds.

Strengths:
-The ablation study seems complete
-Clear improvements over state of the art methods

Weaknesses/improvements:
-The description of the evaluation procedure was a bit vague. Is RANSAC being used to find correspondences? If so, perhaps error bars are necessary to account for variance across multiple runs?
-Make it more clear in the related works about how the proposed method relates to Unsuperpoint. My understanding is that the proposed work is a somewhat incremental improvement over Unsuperpoint.
-Section 3.3 (Score learning) was a bit difficult to follow. I find it better to start by stating the high level goal of the loss function before going into the formulation.
-Captions for Tables 2 and 3 are lacking. At the very least, mention what the numbers being compared are.

Overall, I think the improvements are a bit incremental, but the experiments seem to support the claim that they are beneficial. I had some concerns about the clarity of the paper, and would be willing to raise my rating if addressed.

Post Rebuttal:
The authors have adequately addressed my concerns regarding clarity. I have updated my rating to weak accept in agreement with the other reviews.

**Experience Assessment:**

I have read many papers in this area.

**Review Assessment: Checking Correctness Of Derivations And Theory:**

N/A

**Review Assessment: Checking Correctness Of Experiments:**

I assessed the sensibility of the experiments.

**Review Assessment: Thoroughness In Paper Reading:**

I read the paper thoroughly.

---

> ### Author Response · Authors · 2019-11-14
> **Response to Reviewer #2**
>
> R2: The description of the evaluation procedure was a bit vague. Is RANSAC being used to find correspondences? If so, perhaps error bars are necessary to account for variance across multiple runs?
>
> We thank the reviewer for this suggestion. To address this we added more details about how we compute the homography in the Appendix. Specifically, to estimate the homography, we performed reciprocal descriptor matching and we used OpenCV’s findHomography method with RANSAC, error threshold 3 and a maximum of 5000 iterations.
>
> To capture the variance induced by the RANSAC component during evaluation, we added Table 4 in the Appendix, where we perform 10 evaluation runs with different random seeds and report the mean and standard deviation.
>
> R2: Make it more clear in the related works about how the proposed method relates to Unsuperpoint. My understanding is that the proposed work is a somewhat incremental improvement over Unsuperpoint.
>
> We thank the reviewer for pointing this out. We updated sections pertaining to UnsuperPoint in the related work, and emphasized how our work differs from previous work. We emphasized in the paper that our main contribution is IO-Net, a novel proxy task for the self-supervision of keypoint detection, description and matching. By making the sampling of inlier-outlier sets from point-pair correspondences fully differentiable within the keypoint learning framework, we show that we are able to simultaneously self-supervise keypoint description and improve keypoint matching.
> In addition, with respect to the UnsuperPoint keypoint-network architecture, we propose two modifications that make our approach especially amenable to robust and fine-grained keypoint detection and description. We modify the keypoint location head to detect keypoints outside their corresponding cell-boundaries, allowing for improved keypoint matching especially at cell-boundaries. We also further improve fine-grained keypoint descriptor performance by taking advantage of efficient sub-pixel convolutions to upsample the descriptor feature-maps to a higher operating resolution. We have updated the text to clarify these details further.
>
>
> R2: Section 3.3 (Score learning) was a bit difficult to follow. I find it better to start by stating the high level goal of the loss function before going into the formulation.
>
> We have updated Section 3.3 to include a high-level description of the loss function, before moving on to the formulation. Our main goal is two-fold: (i) we want to ensure that feature-pairs have consistent scores, and (ii) the network should learn that good keypoints are the ones with low feature point distance.
>
> R2: Captions for Tables 2 and 3 are lacking. At the very least, mention what the numbers being compared are.
>
> We have updated these captions with clear explanations of what is being discussed in each table. More specifically, in Table 2 we show that our proposed method outperforms all other listed traditional and learned feature methods in repeatability, and for localization we outperform UnsuperPoint for higher resolution images. In Table 3, we show that our proposed method outperforms all listed methods for different pixel distance thresholds, except for SIFT in one single instance (the Correctness-1 metric).
>
> R2: Overall, I think the improvements are a bit incremental, but the experiments seem to support the claim that they are beneficial. I had some concerns about the clarity of the paper, and would be willing to raise my rating if addressed.
>
> We thank the reviewer for the feedback, and hope that the improvements to the paper's clarity and contributions sufficiently addresses these concerns.

---

### Author Response · Authors · 2019-11-14
**Response to Reviewers**

First of all, we would like to thank all the reviewers for their feedback and useful suggestions, and we are glad to see the recognition of the novelty of our work, relevance, and state-of-the-art results. As the reviewers have pointed out, we acknowledge that our paper would certainly benefit from a clearer and more detailed technical presentation. To this end, we have updated the manuscript in the following ways:

* Clarified the description of our contributions in the related work, particularly in regards to UnsuperPoint.
* Added a detailed description of our neural network architecture in the Appendix.
* Improved the figures/tables and associated captions.
* As requested by the reviewers, we conducted new experiments to quantify (i) the standard deviation of our homography estimation (ii) additional qualitative and quantitative experiments on HPatches (viewpoint, illumination and specific sequences).
* Fixed minor corrections and typos that were accurately pointed out by the reviewers.

To facilitate this rebuttal process, we would like to suggest referring to our updated manuscript and detailed comments on each of the reviewers’ points, that were addressed individually.

---

### Decision · Program_Chairs · 2019-12-19

**Decision:**

Accept (Poster)

**Comment:**

This paper proposes a solid (if somewhat incremental) improvement on an interesting and well-studied problem. I suggest accepting it.